# Integrated Approach for Human Wellbeing and Environmental Assessment Based on a Wearable IoT System: A Pilot Case Study in Singapore

**DOI:** 10.3390/s24186126

**Published:** 2024-09-22

**Authors:** Francesco Salamone, Sergio Sibilio, Massimiliano Masullo

**Affiliations:** 1Construction Technologies Institute, National Research Council of Italy (ITC-CNR), Via Lombardia, 49, San Giuliano Milanese, 20098 Milano, Italy; sergio.sibilio@unicampania.it; 2Department of Architecture and Industrial Design, Università degli Studi della Campania “Luigi Vanvitelli”, Via San Lorenzo, Abazia di San Lorenzo, 81031 Aversa, Italy; massimiliano.masullo@unicampania.it

**Keywords:** IEQ, OEQ, wearable, multi-domain environmental monitoring system, IoT, machine learning

## Abstract

This study presents the results of the practical application of the first prototype of WEMoS, the Wearable Environmental Monitoring System, in a real case study in Singapore, along with two other wearables, a smart wristband to monitor physiological data and a smartwatch with an application (Cozie) used to acquire users’ feedback. The main objective of this study is to present a new procedure to assess users’ perceptions of the environmental quality by taking into account a multi-domain approach, considering all four environmental domains (thermal, visual, acoustic, and air quality) through a complete wearable system when users are immersed in their familiar environment. This enables an alternative to laboratory tests where the participants are in unfamiliar spaces. We analysed seven-day data in Singapore using a descriptive and predictive approach. We have found that it is possible to use a complete wearable system and apply it in real-world contexts. The WEMoS data, combined with physiology and user feedback, identify the key comfort features. The transition from short-term laboratory analysis to long-term real-world context using wearables enables the prediction of overall comfort perception in a new way that considers all potentially influential factors of the environment in which the user is immersed. This system could help us understand the effects of exposure to different environmental stimuli thus allowing us to consider the complex interaction of multi-domains on the user’s perception and find out how various spaces, both indoor and outdoor, can affect our perception of IEQ.

## 1. Introduction

In recent years, Indoor Environmental Quality (IEQ) has attracted the attention of several researchers [1], as the quality of indoor spaces affects the well-being, health, and productivity of occupants. In the scientific literature, there are several case studies dealing with the assessment of specific environmental domains of IEQ among thermal comfort, visual comfort, acoustic quality, and Indoor Air Quality (IAQ) [2]. Several other papers have analysed the existing literature to understand the human responses to multi-environmental stimuli. Schweiker et al. [3] provided a comprehensive review of the influences of multi-domain stimuli on occupant perception and behaviour based on field and laboratory studies. Torresin et al. [4] and Berger et al. [5] focused on laboratory studies that examined human perception and performance when participants were exposed to at least two different environmental stimuli. Chinazzo et al. [6] pointed out several shortcomings in the design, use, and documentation of multi-domain studies and emphasised the need for quality improvements in future multi-domain research and the consolidation of our knowledge on multi-domain exposures instead of the single-domain guidelines that currently dominate. This work shows a weakness in the results due to an unclear tendency for interaction between the domains. This finding could be explained precisely by the lack of any consolidated procedure in the analysis of multi-domain studies. Schweiker also refers to the complexity of the interactions due to many influencing factors and/or the heterogeneity of the topics under investigation. Other studies [7,8,9,10] have confirmed this aspect and have tried to find the reasons for it, such as the lack of interdisciplinary research that overcomes the dichotomy between building physics and psychophysical user perception. According to [11], such research should involve a variety of people with heterogeneous scientific backgrounds, ranging from engineering to medicine. Simultaneous research of objective (engineering) and subjective (psychological–perceptual) matrices to standardise procedures requires dealing with qualitative and quantitative data. According to Choy [12], there is no best approach between the qualitative or quantitative-oriented research methods, as both types of research have their strengths and weaknesses. Fransson et al. [13] investigated the relative usefulness of subjective (rating scales) and objective indicators of perceived comfort in indoor hospital environments and concluded that subjective sensory ratings were significantly better than objective indicators in predicting overall rated comfort in indoor environments.

In studying the environmental quality as perceived by the occupants, there are some other specific considerations to consider. For instance, the individuals could modify their behaviour, performance, or preference in response to being observed or participating in an experiment. Although this effect, called the Hawthorne effect [14], has not been well analysed in tests within living laboratories, it could have a “disruptive influence” [15] on the feedback of testers involved in experiments in living laboratories.

Another important issue in thermal comfort assessment is to take into account the time of the day [16], since thermal alliesthesia, i.e., the influence of our perception of temperature due to our internal physiological state and the context in which we experience it, significantly affects thermal comfort [17]. A time-based visual comfort assessment is also an aspect that can be considered [18]. In this specific case, even though the concept of alliesthesia does not have a direct equivalent specifically designated for visual comfort perception, the way we perceive visual stimuli is not solely determined by the objective properties of the visual input and of the surrounding environment, but it is also influenced by various internal factors, including physiological [19] and psychological or emotional states [20]. At the same time, while there is no extensive literature that specifically uses the term “acoustic alliesthesia”, the broader concepts of emotion, physiology, context, and culture that influence acoustic quality perception are well established in fields such as cognitive neuroscience, music psychology, and psychoacoustics [21]. The same applies to air quality, because the perception of air pollution is also linked to people’s attitudes, feelings, and psychological aspects [22], and to certain contexts [23].

In summary, the dynamic perception of comfort and wellbeing refers to the understanding that an individual’s perception of the environmental quality (in all four domains) and of well-being is not static but can change in response to various environmental, physiological, and psychological factors over time. This concept recognises the dynamic nature of the human experience and the continuous interaction between individuals and their surroundings. Furthermore, this dynamism in assessing the perception of wellbeing is not really considered by the scientific literature because the classical approach considers an adaptation time that in most of the laboratory studies is of 30 min [6] and then an elapsed time for the test that is about 200 min on average with a great variability, between 3 min and 1 day and a half, depending on the main objective of the research. This is clearly a limitation due to the use of facilities that require monitoring, both for maintaining specific environmental conditions and for accompanying the test that suffers from a classical approach in which all sensors are in a given location and participants follow each other in the tests, away from their daily environment.

Starting from this premise, we ask the following question: Is it possible to use an alternative to the “consolidated” approach of analysing data collected for a short time in the laboratory to an approach where one can imagine creating a wearable-based framework where it is possible to follow the user as they move while monitoring objective environmental variables, subjective feedback, and individual physiological parameters?

To answer this question, we designed and developed a pilot study and applied it in a real-life scenario to assess the feasibility of a complete wearable system to capture environmental variables, user feedback, and physiological parameters in a real-life scenario. While there is already available hardware that can be used for the physiological parameters and user feedback, for the monitoring of the environmental variables in a multi-domain perspective, we have not found [24] a completely wearable system that can be used for this purpose. For this reason, we have developed the WEMoS (Wearable Environmental Monitoring System) device. This is the result of a three-year research study. And in defining this wearable-based framework based on WEMoS and other available solutions described in the next section, it was logical to start with a pilot study, as this is usually used to check whether the design of the test methodology, i.e., the application of the wearable-based framework in a real context, really works and whether the method can be applied, so that one can move on to the main study [25]. With this approach, we can investigate and show how the available monitoring data can be used in a holistic predictive black-box analysis.

The article presents the results of this research study and is divided into the following sections: Section 2 describes the materials and methods used to conduct the tests, Section 3 describes the results in terms of descriptive and predictive approach, Section 4 describes the discussion and Section 5 reports the conclusions.

## 2. Materials and Methods

The pilot test was conducted in May 2023. An indoor space in Singapore was considered as the case study. No changes in the setup of the indoor space in terms of air temperature, ventilation rate, illuminance level, were performed during the pilot test thus allowing us to maintain familiarity with that environment. Five different users took turns in the test (for more details, please read Section 2.1). They read the informed consent and accepted it before expressing their willingness to perform the test. For further details, refer to Section 2.1. Depending on the availability of participants, a testing schedule was established. During the pilot test, each participant wore a completely wearable system. It consists of the following three main parts: a smartwatch with the Cozie app for subjective feedback (Section 2.2), the wristband for physiological monitoring (Section 2.3), and the WEMoS for environmental monitoring (Section 2.4). During the test, participants were asked to give their feedback every 30 min. They were free to move outdoors. We only asked that, before moving outdoors, they provide feedback about their comfort perception, even if half an hour had not elapsed. Also, when they returned indoors, they were once again asked to provide feedback in terms of comfort perception.

### 2.1. Case Study and Testers

During the seven-day test conducted between 22 May and 30 May 2023, an indoor open-plan office/study building in Singapore was considered as the case study. It has an air volume of 545 m^3^, not including the air volume of a small meeting room of 116 m^3^ that is also available and is equipped with a hybrid cooling system with a ceiling fan (Figure 1).

Five different users took turns in the test. They had a level of education higher than a Bachelor’s degree, were in the age range of 22–38 years old, weighed 58–87 kg, and were in the range of 163–182 cm in height. Table 1 shows the details (gender, test time, and use of Empatica E4 during the test) of each participant identified with a pseudonym.

Since they were wearing a fully wearable system during the test, they could move freely outdoors or into other rooms, without any restrictions.

Although the research involved human subjects, following the indication of the Human Biomedical Research Act 51 of the Ministry of Health of Singapore, and of other relevant worldwide ethical guidelines/regulations (see: National Health and Medical Research Council National Statement on Ethical Conduct in Human Research 2007—Australia 52; 45 CFR 46 Section 104 and Department of Health and Human Services, Office for Human Research Protections, Human Subject Regulations Decision Charts 54—United States; Central Committee on Research Involving Humans, The Netherlands), as our study involved a “minimal risk” to subjects and included only behavioural research, such as surveys or passive observation, it was deemed exempt from the ethics review.

At the same time, all the standards for the research were adhered to, and the research objectives were explained to the participants. Following the principles of the Declaration of Helsinki, participants gave their informed consent before participating in the test. They were informed that participation was voluntary and that they could interrupt or withdraw from the test at any time. To protect the confidentiality of the participants, the data were pseudonymized.

### 2.2. Smartwatch with Cozie App

To acquire user feedback about the comfort perception related to the four most significant environmental factors, we used an Apple smartwatch with the Cozie app installed, defined as “a framework to collect feedback from building occupants in real time” [27,28,29]. The Cozie app is available for the iOS smartwatch and Fitbit and is usually used for research purposes to conduct micro-surveys, which are primarily used to collect human feedback. In this case, a development version of Cozie was used, the Cozie v3, and a custom watch survey was made for the test (Figure 2). The Cozie app was used to ask participants in the test to provide feedback about, in particular, location, activity performed, and perception of air quality, visual comfort, thermal comfort, acoustics, and overall comfort perception. The rating is based on a unipolar 5-point visual categorical scale.

Using a perception question in the form of “How do you perceive the AIR QUALITY/VISUAL/THERMAL/ACOUSTIC/GENERAL COMFORT conditions since your last feedback?”, we asked participants to give us feedback on their average perception since their last feedback. In this way, we could use the point-by-point feedback and compile the time period since the last feedback with the average feedback provided so that we could consider a completely subjective data column without “NAN” in a complete database containing environmental and physiological data as well.

### 2.3. Physiological Monitoring Device

The physiological monitoring device used was the Empatica E4, a wristband class II medical device according to the 93/42/EEC Directive [31] equipped with the following sensors:a photoplethysmography (PPG) sensor for the detection of the heart rate (HR);an electrodermal activity (EDA) sensor;an infrared thermopile;a 3-axis accelerometer.

The Empatica E4 wristband was chosen because it was the most practical solution for the purpose of the research. Table 2 reports the characteristics of the sensors used for the acquisition of biometric data.

### 2.4. WEMoS Prototype

The WEMoS consists of two parts, the first of which can be attached to the belt at waist level and the second to the head (Figure 3). In the wearable part at the waist level, the following are monitored: air temperature and relative humidity. In this case, in accordance with the results of this study [32], we chose the DHT22 thermohygrometer [33] and, in agreement with the results of [34], we considered two measurement points, at different distances from the human body, one sensor close to the body and the other at least 8 cm away, to determine if and how the standard thermal comfort assessment differs from the user’s personal perception and whether spatial proximity might also play a role. In our view, this is an important development because none of the studies that have used thermohygrometers in the development of wearables, as shown in [24], have considered the distance to the human body and whether/how much the human thermal plume might affect the measurement of environmental variables (air temperature and relative humidity). The wearable part at the waist level also contains a Senseair K30 carbon dioxide concentration sensor [35], an Adafruit PMSA003I particulate matter concentration sensor [36] (for monitoring PM1, PM2.5 and PM10 particle size), and a Modern Device hot-wire anemometer [37]. All these sensors are connected to an Arduino Micro Board [38] that manages all the information and sends all the data to a Bluetooth Low Energy (BLE) module that transmits the data to a Raspberry Pi 3A+ [39] at the head level, which has a data logging function and stores all the data collected at waist level as well as the data from the other sensors at the head level, namely the luminance detected by a wide-angle camera [40] in front of the user, visible spectrum (from which we derive: α-opic irradiance, α-opic Equivalent Daylight Illuminance, α-opic Efficacy of Luminous Radiation, α-opic Efficacy Ratio, Illuminance at eye level—E_lx, tristimulus values (XYZ), Correlated Color Temperature—CCT, Color Rendering Index—CRI, Melanopic Equivalent Daylight Illuminance—MEDI, Circadian stimulus values—CS, Circadian Light values—Cla), and some kind of mean radiant temperature, measured by 4 infrared modules MLX90640 with 110° FoV [41]. WEMoS was realized in the following two different versions: one with an integrated power supply (in this case, a Li-ion battery charging module with a 2300 mAh Li-ion battery connected to the device at the waist level and a PiSugar2 Plus module [42] with a 5000 mAh battery capacity connected at the head level); another, as expected in this case when the device is used for air travel where it is useful to disconnect the battery, is powered by two external 10,000 mAh power banks for both devices at the waist level and at the head level. For all the technical details, please refer to the references provided. All the electronics were housed in a 3D-printed case designed with Rhinoceros 7 (Figure 4), in line with the Do-It-Yourself (DIY) philosophy [43]. The weight of the entire WEMoS is about 600 g (581 g in the case of the integrated power supply and 665 g in the case of the external power supply with the two power banks).

### 2.5. Data Preparation

WEMoS provides environmental data at 5 min intervals. For this reason, physiological data were also averaged considering 5 min. The average perceptual responses for each epoch were also reported in the same database. Most of the columns in the response database, especially those related to comfort perception of the four environmental domains and overall comfort, are of the object type, i.e., they contain strings. These variables are categorical and are stored as text values. In the encoding phase, the responses are converted into numeric values for further processing, since many machine learning algorithms (ML) can support categorical labels without further modification, but many others cannot. After filtering columns used only for white-box assessment and containing different NAN values, the overall database contains 403 non-zero values for all features except those related to physiological data (267 non-zero values) because, as indicated in Table 2, not all participants wore the Empatica E4 during the test.

## 3. Results

### 3.1. Descriptive Analysis on the Monitored Environmental Variables

Below are some figures showing the line plots of several environmental variables monitored by WEMoS with a frequency of 5 min. Figure 5 shows the environmental variables that were monitored in relation to the thermal domain.

As shown previously in [34], T_0_, the body temperature near the waist, is higher than T_8_, the temperature measured eight centimetres from the human body.

Figure 6 shows the trends of variables related to the air quality domain. In particular, the figure considers CO_2_ and PM1, PM2.5, and PM10 concentrations measured at the waist level.

No significant changes in CO_2_ concentration were recorded during the test. PM levels tended to be highest when the users went outdoors; this is likely due to the nearby road, which is also used by heavy vehicles from the port.

Figure 7 shows the trends of the variables related to the visual area. The figure considered the E_lx, the vertical illuminance at eye level, and the CCT in [k], the Correlated Colour Temperature.

While there is no significant difference in CCT because the light used indoors is mainly daylight passing through the fully glazed walls bordering the case study, during the test hours on the different days, apart from Anga Test 1 on 22, 24, and 29 May, and Anga Test 5 on May 30, the illuminance levels (E_lx) are very different between the time indoors and the time outdoors in most cases.

Figure 8 shows the trends of the variables related to the acoustic domain. In this case, we considered both A- and C-weighted sound equivalent levels for the two microphones (L and R).

Regarding subjective feedback, we showed the aggregated value during the test periods in terms of perception of four environmental domains and the overall comfort (Figure 9).

For all four environmental domains, the average overall perception is “comfortable”.

### 3.2. Predictive Data Analysis

Due to the complexity of the interaction between the environment and user perceptions, we considered a dataset that includes both objective data, as monitored or calculated from WEMOS, and subjective data (user feedback acquired through Cozie app and physiological data monitored through Empatica E4). In this sense, we have several environmental variables related to several environmental domains, some user feedback related to comfort perception in each of the four domains (thermal, acoustic, visual, and air quality), and the general comfort perception and some columns related to the physiological data. We used the overall comfort perception as a target feature.

First, we used the nonparametric Spearman’s test to check the correlation between the variables or, in other words, the features (Figure 10).

Some dependencies between features can be seen by the red or green colours. Therefore, we could exclude these features. Based on the above features, we only considered the following features in the black-box model used to predict overall comfort perception (‘q_general_comfort_condition’):A Boolean value defining the position of the user (In1_Out0’);The CO_2_ concentration in ppm (‘CO2_ppm’);The PM1 concentration in μg/m^3^ (‘PM1’);The relative humidity in % at 8 cm from the body (‘RH_8_%’);The air temperature in °C at 8 cm from the human body (‘T_8_Â°C’);The air velocity in m/s (‘Va_m/s’);The Mean Radiant Temperature in °C derived as an average value from 4 IR modules (‘MRT_°C’);The illuminance in lx (‘E_lx’);The Correlated Colour Temperature in K (CCT);The equivalent continuous sound level (A-weighted) of R channel in dB (‘LAeq_R’);Heart rate variability in ms (‘HRV’);Electrodermal Activity in μSiemens (‘EDA’);Blood Volume Pulse in μV (‘BVP’);Skin temperature at wrist level in °C (‘TEMP’);Data of x acceleration in the range [−2 g, 2 g] (‘ACC_X’);Data of y acceleration in the range [−2 g, 2 g] (‘ACC_Y’);Data of z acceleration in the range [−2 g, 2 g] (‘ACC_Z’);Data of overall acceleration in the range [−2 g, 2 g] (‘ACC_Overall’);Identification string assigned to each participant (‘id_participant’);Answer related to the question “wearing ear/headphones?” (‘q_earphones’);Answer related to the question “how do you perceive the VISUAL conditions since your last feedback?” (‘q_visual_condition’);Answer related to the question “how do you perceive the THERMAL conditions since your last feedback?” (‘q_thermal_condition’);Answer related to the question “how do you perceive the AIR QUALITY since your last feedback?” (‘q_air_quality_condition’);Answer related to the question “how do you perceive the ACOUSTIC conditions since your last feedback?” (‘q_acoustic_condition’).

In more detail, three different lists of features were considered (Table 3).

To select the right estimator, we used the flowchart of scikit-learn [44] and identified Linear Support Vector Classifier (LSCV) [45], Support Vector Classifier (SVC) [46], Kneighbors Classifier (KNN) [47], Random Forest Classifier (RF) [48], Gradient Boosting Classifier (GBC) [49], and Extra Trees Classifier—ETC [50] as the best algorithms to use in this case. In this case, we used the metric “F1-score” to evaluate the different algorithms. It was defined as the weighted average of precision (defined as a measure of a classifier’s exactness) and recall (considered as the completeness of the classifier). Stratified k-fold cross-validation (number of splits = 10) was used to evaluate the performance of the different algorithms. Table 4 shows the range of tuned hyperparameters considered for each algorithm. Among the different techniques available for tuning hyperparameters, the GridSearchCV method [51] was used. This method starts with the definition of a search space grid. The grid consists of selected hyperparameter names and values, and the grid search exhaustively searches for the best combination of these given values.

Figure 11 shows the average F1-score and standard deviation for the different models and features considered.

The RF model slightly outperforms the GBC model, and it has the highest average F1-score considering list 2 of the features, without physiological data (0.983 ± 0.017). The result in terms of F1-score on unseen validation data is 1.0, in line with the results obtained with training data.

For the RF-trained model and list 2 of the considered features, it was possible to explain the results of the SVM-based model by applying the SHAPley additive explanations (SHAP) library, a game-theoretic approach that allowed us to identify the importance of all selected features [52,53]. Figure 12 shows the distribution of the impact of each feature on the model output.

The plot variables are classified in descending order of importance. The points, each of which represents a prediction, are distributed along the horizontal axis; the farther they are from the centre of the X-axis (SHAP value = 0.0, i.e., no effect on the model output), the greater the impact. The positive SHAP values of the points on the right side represent a positive impact on the quality score, while the negative points are associated with an antagonistic impact. Figure 12 shows that, among the objective environmental data, CCT, MRT, and CO_2_ have the largest positive impacts, while among the subjective features, q_thermal_condition is the most important in predicting overall comfort satisfaction.

## 4. Discussion

The pilot test conducted in Singapore demonstrates the practical benefits of using a fully wearable system. Using the data collected, it is possible to take a descriptive approach thus allowing us to understand the trends of environmental variables in the participants’ surroundings in real time. As described above, we can also understand how the environmental variables change when the users move from an indoor to outdoor space. With this aspect, we can add a discussion point that has not really been considered in the literature but which we believe could be very helpful in forming new research directions. Consider the following Figure 13 as an example. In Figure 13a, you can see a profile of air temperature taken with a “nearable” device [54], i.e., a smart monitoring device that is closer to the user. This is a segmented trend because the air temperature data are displayed when the user is close to the monitoring device; from a human-centered perspective, only these data are very useful to understand the possible correlations between comfort perception and environmental status. In Figure 13b, we can track the user when they are moving and thus understand what environmental influences the user is exposed to. Although no one has yet been able to quantify this, it is logical — and by this we do not think we are introducing speculative thinking — that a moment in real life when the user is outdoors (orange arrows in Figure 13b) could play a role in the assessment of thermal comfort.

The use of a wearable-based framework such as the one described here, which is completely wearable and whose feasibility we have validated through this pilot test, could provide some help in quantifying how the dynamic perception of comfort and wellbeing, already discussed in introduction, may change in response to the various environmental stimuli, both indoor and outdoor, to which a user is subjected throughout the day. In the context of a comprehensive wearable system, the considerations for defining comfort go beyond the cross-modal and combined aspects of the four environmental factors. It is also about investigating how outdoor conditions can influence the perception of comfort. Consequently, data collection on human interaction with the environment should include both indoor and outdoor data thus allowing for their contribution to users’ perception of comfort to be quantified.

Furthermore, since we can monitor head movement, luminance tracking is possible with the wearable device, a capability which is missing from a nearable. The same can be considered for acoustic monitoring with the two microphones closest to the user’s ears. This holistic approach to environmental monitoring, covering all four environmental domains comprehensively, allows us to potentially broaden our understanding in relation to the cross-modal and combined aspects mentioned in the introduction.

The DIY approach to device design is proving to be cost-effective and facilitates the proliferation of affordable solutions for capturing information about the user’s environment, which is in line with the Internet of Things (IoT) paradigm. DIY and IoT represent the third wave of the information industry revolution, following computers and the internet. As we move into the era of Smart IoT, the abundant data from “Smart Things” are being used to train machine learning algorithms and develop AI-based applications. Notwithstanding these developments, we have shown that it is possible to apply the WEMoS to track various environmental variables related to the four key environmental factors. By integrating the WEMoS into a framework with other existing wearable devices, we can create a comprehensive database of environmental, physiological, and user feedback data. The application of this methodological approach can be facilitated by incorporating DIY and IoT principles, providing insights into how different environmental stimuli—whether combined or cross-modal, indoor or outdoor—influence the perception of human well-being.

### 4.1. Limitations of the Proposed Study

As is logical in a pilot test, the results may be affected by some limitations due to the small number of users involved in the test or the single case study. Both are aspects that may impose some limitations, as described in more detail below.

While the proposed study presents a comprehensive wearable-based framework capable of monitoring the environmental, physiological, and subjective comfort parameters, it is not able to comprehensively describe the interaction between these different data sources, which is crucial given the complex interactions between these factors. In the field of environmental and comfort studies, understanding how these parameters influence each other is critical to drawing meaningful conclusions. For example, physiological responses do not always coincide with subjective perceptions of comfort, and environmental conditions may affect individuals differently depending on physiological or psychological factors.

Another aspect that we did not investigate, but could be analysed in depth, is the possible recall bias that arises from asking participants to reconsider their feedback since the last input. This is particularly problematic when capturing subjective experiences such as well-being, as the recall bias may lead participants to misrepresent the intensity or duration of their feelings. Recall bias can be greater when the study participant has poorer recall in general (not in our case) and when asked about events over a longer period (in our case, we asked about a short period of time, about 30 min of experience).

To be successful in the real world, wearable systems need to be user-friendly, comfortable and integrate seamlessly into daily routines without causing discomfort or requiring constant attention.

For these reasons, we would like to emphasize that the results presented here cannot be generalized. We would like to emphasize that our study is a pilot test focusing on the feasibility of the presented wearable-based framework and how the monitored data can be analysed considering a descriptive and predictive approach. All the above limitations of the proposed pilot test could be considered in a wider application of the proposed wearable-based framework.

### 4.2. Further Implications Regarding the Application of the Wearable-Based Framework in the Real World

The wearable-based framework presented here, if applied in real-world contexts, could be relevant because it could allow for the following:provide continuous, individualized monitoring of physiological, environmental, and subjective comfort parameters, enabling personalized interventions in healthcare, workplace ergonomics, and daily well-being by allowing real-time adjustments based on personal comfort or health metrics.enable a more holistic understanding of a person’s comfort or health status, thanks to the possibility of using multiple data sources (environmental, physiological, and subjective data). This can be used in various areas such as urban planning, building design, and occupational health, where data-driven insights can help optimize the environment for human comfort and performance.provide a wealth of data that, if used on a larger scale, can help researchers and policymakers better understand population trends in comfort and health. This could serve as a basis for public health strategies, workplace regulations, and even product design to improve human wellbeing in different areas.

## 5. Conclusions

In conclusion, we found the following:The DIY approach enables the construction of a wearable device to assess environmental monitoring in a descriptive way.It is applicable in real-world contexts for longer test periods than those carried on in the laboratory.The environmental data collected with the WEMoS can be merged with physiological information and user feedback, helping to identify key features that are important for defining the overall perception of comfort.The potential of this wearable-based framework in the real world is enormous, ranging from improving personal health and comfort to influencing environmental and health policy on a large scale.

Beyond the technical aspects on individual sensors used in the development of WEMoS, the following general limitations of the proposed approach arise:Each element of the wearable system for monitoring environmental variables should undergo instrumental verification before it can be used.An initial training phase related to the use of the devices is required.The architecture of the wearable-based framework as used in the test performed in Singapore could be more integrated, so that all information could converge to a single database.As is well known, predictive models do not allow for good interpretability of results, so it is necessary to always accompany a descriptive phase of the data monitored during the test.

## Figures and Tables

**Figure 1 sensors-24-06126-f001:**
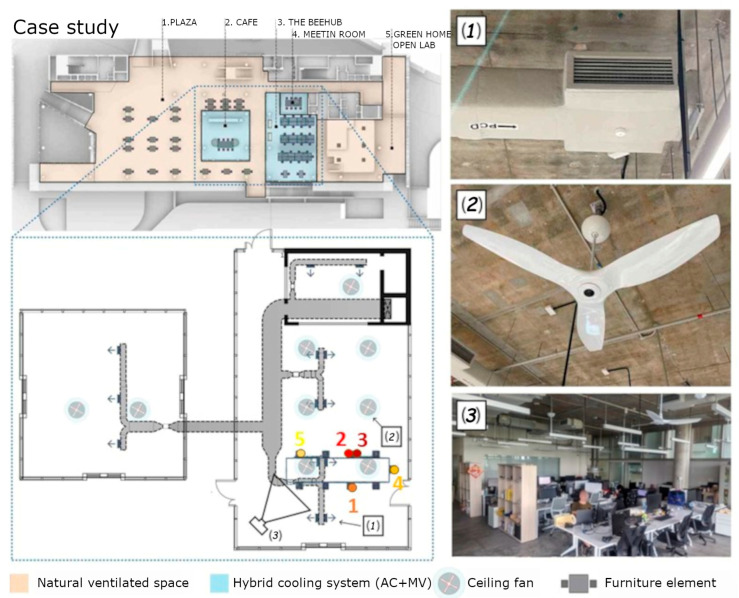
Case study: details of the floor of the building where the indoor space used for the test is located with the position of testers (1–5) and some photos of hybrid cooling system (**1**), the ceiling fan (**2**) and the overall space with furniture (**3**). Original Figure available in [26]. License provided by Elsevier (License number 5861251407928).

**Figure 2 sensors-24-06126-f002:**
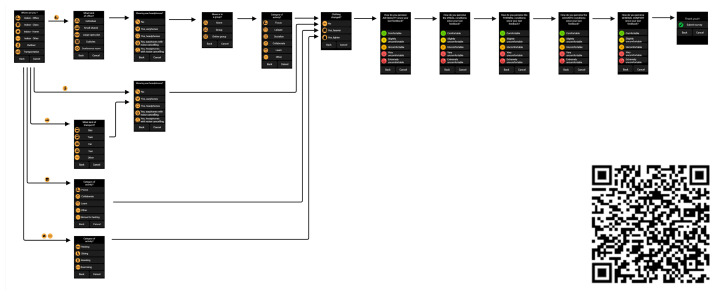
Details of the flowchart of the Cozie app questions. Please see the QR code or [30] to navigate through the flowchart.

**Figure 3 sensors-24-06126-f003:**
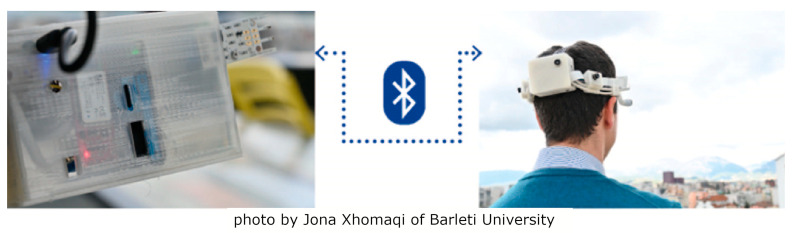
Fully wearable system with a detail of WEMOS (Wearable Environmental Monitoring System) used during the test experiment.

**Figure 4 sensors-24-06126-f004:**
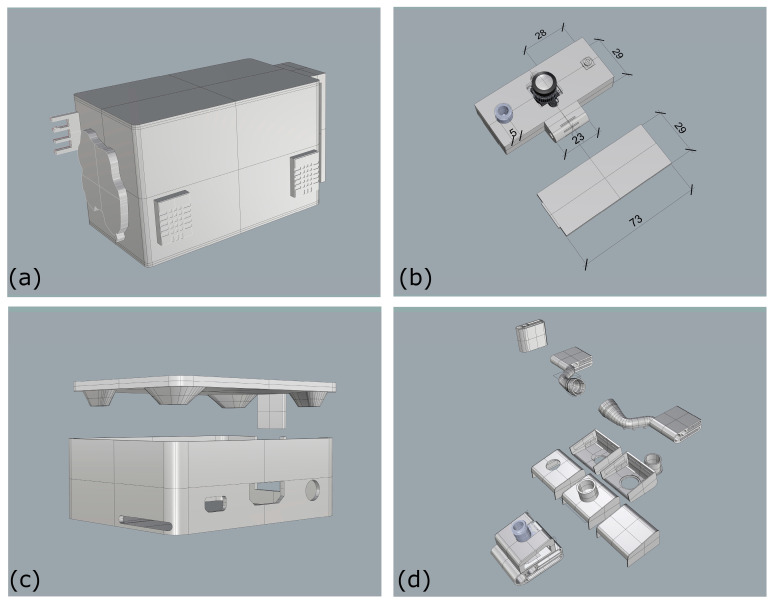
WEMoS’ parts and cases as designed with Rhinoceros: (**a**) part at waist level; (**b**) for wide-angle camera, one IR sensor, and the spectroradiometer (measures in millimeters); (**c**) the case for the Raspberry Pi 3A+; (**d**) different clips for the two microphones, the cables, and the IR sensors.

**Figure 5 sensors-24-06126-f005:**
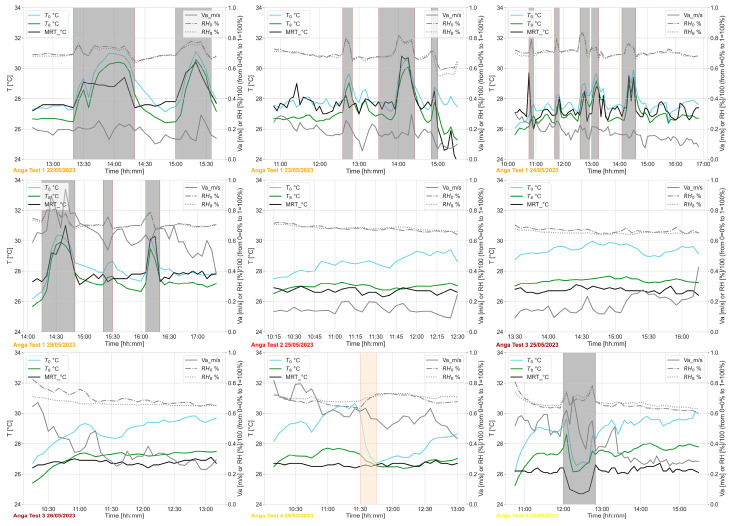
Thermal domain–T_0_ and RH_0_ (air temperature °C and relative humidity % near to the user), T_8_ (air temperature °C and relative humidity % at 8 cm from the body), Va (air velocity in m/s) and MRT (Mean Radiant Temperature in °C) as monitored during the periods of test. The grey areas refer to the times when the user is outdoors or in other rooms, while the orange area refers to the time when the WEMoS was not worn by the user.

**Figure 6 sensors-24-06126-f006:**
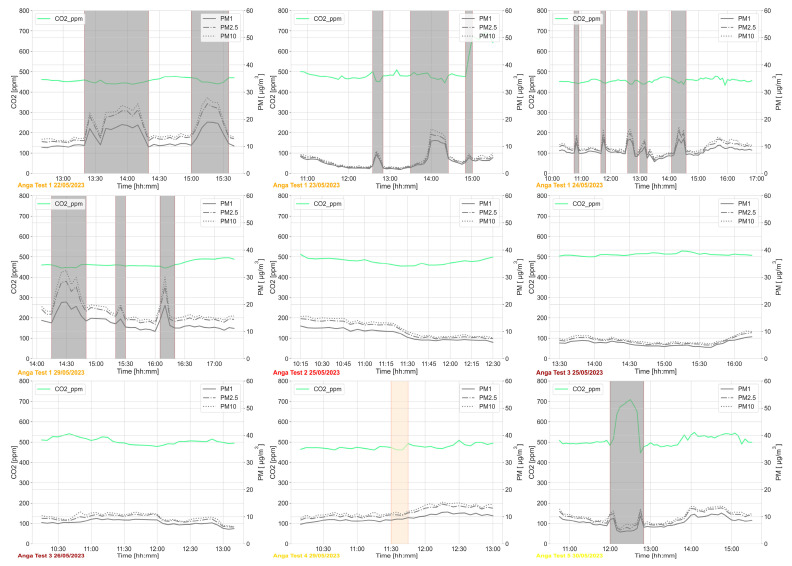
Air quality domain—CO_2_ and PM concentrations monitored during the periods of test. The grey areas refer to the times when the user is outdoors or in other rooms, while the orange area refers to the time when the WEMoS was not worn by the user.

**Figure 7 sensors-24-06126-f007:**
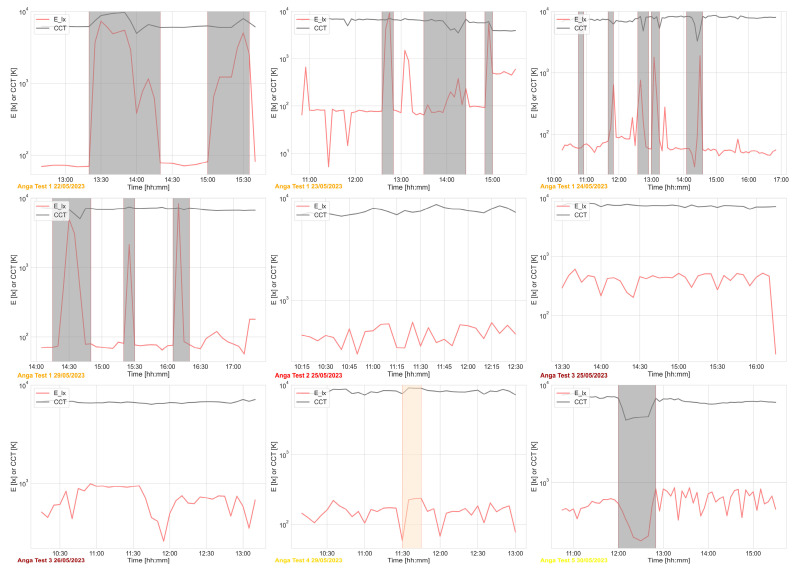
Visual domain—E_lx (illuminance at eye level in lx) and CCT (correlated colour temperature in K) monitored during the periods of test. The grey areas refer to the times when the user is outdoors or in other rooms, while the orange area refers to the time when the WEMoS was not worn by the user.

**Figure 8 sensors-24-06126-f008:**
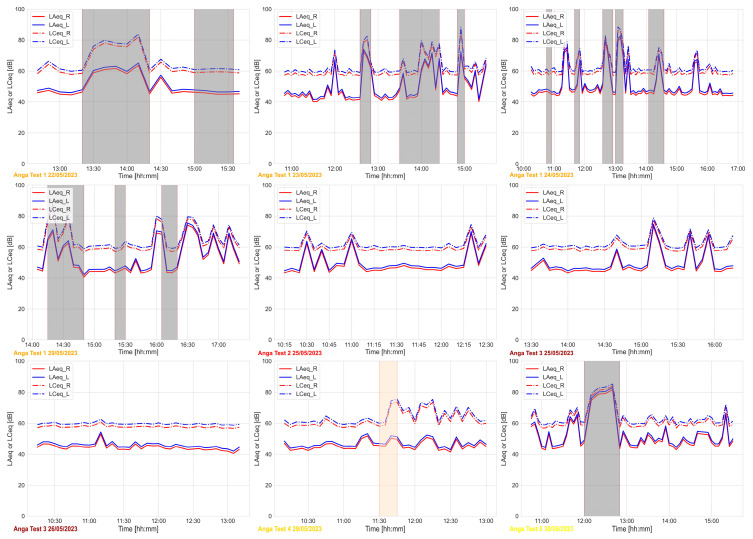
Acoustic domain—LAeq_R and LAeq_L (A-weighted continuous sound equivalent level for R and L channel) LCeq_R and LCeq_L (C-weighted continuous sound equivalent level for R and L channel) monitored during the test periods. The grey areas refer to the times when the user is outdoors or in other rooms, while the orange area refers to the time when the WEMoS was not worn by the user.

**Figure 9 sensors-24-06126-f009:**
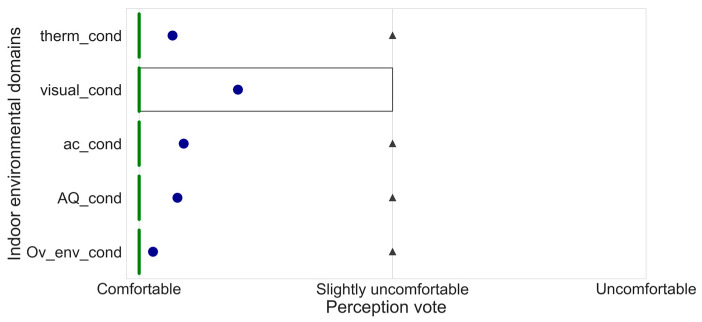
Feedback on satisfaction. The blue dot indicates the mean values, the black triangle the fliers, and the green lines indicates the median. Thermal perception = “therm_cond”, Visual perception = “visual_cond”, Acoustic perception = “ac_cond”, Air Quality perception = “AQ_cond”, Overall Environmental perception = “Ov_env_cond”.

**Figure 10 sensors-24-06126-f010:**
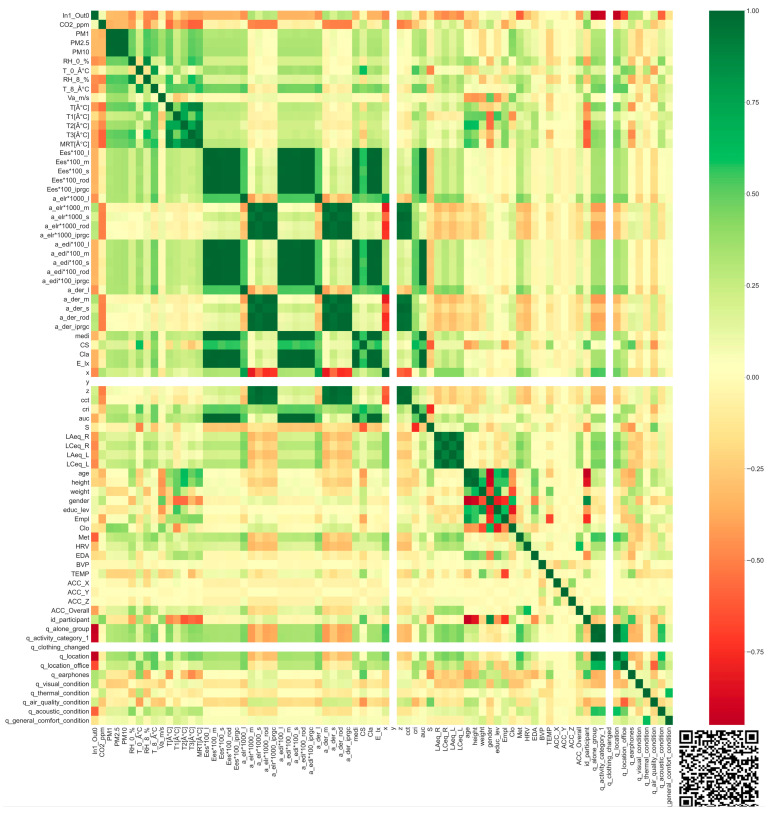
Spearman’s correlation matrix. It measures the strength of association between two variables in a single value between −1 and +1. QR code links to the high-resolution image with the values reported in each cell.

**Figure 11 sensors-24-06126-f011:**
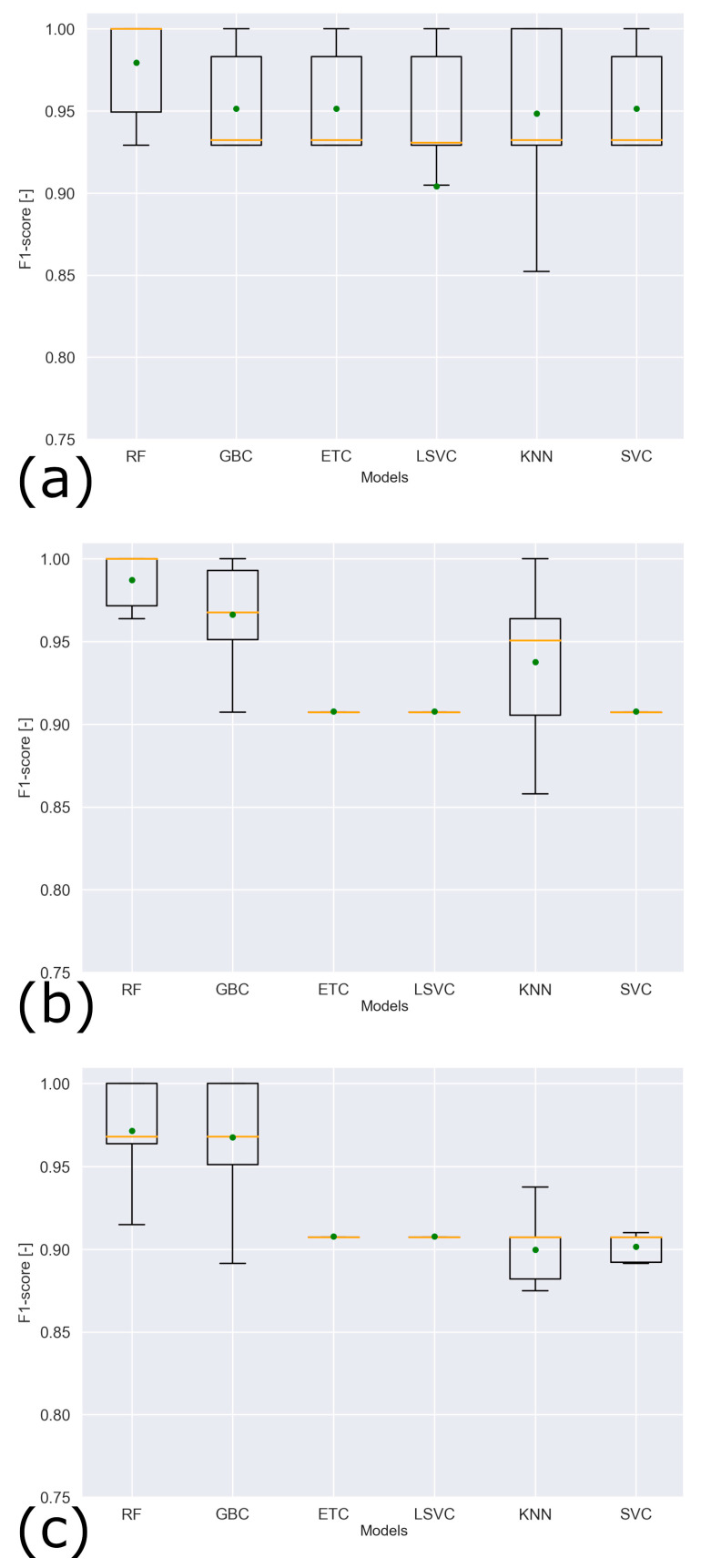
F1-scores for the different models and features considered: (**a**) list 1; (**b**) list 2; (**c**) list 3. The green dot indicates the mean value, and the yellow line indicates the median.

**Figure 12 sensors-24-06126-f012:**
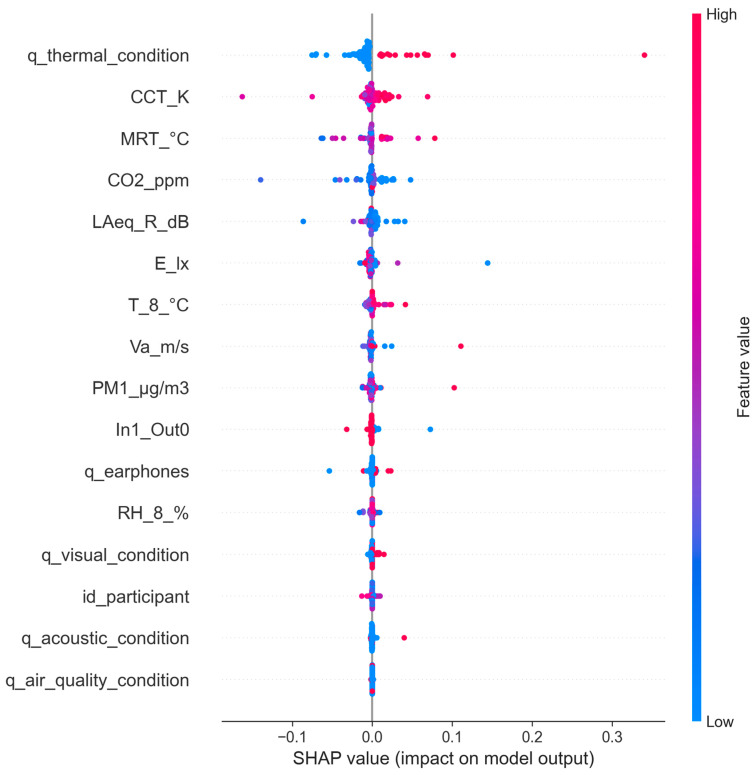
SHAP summary plot: High feature values in red; low features values in blue.

**Figure 13 sensors-24-06126-f013:**
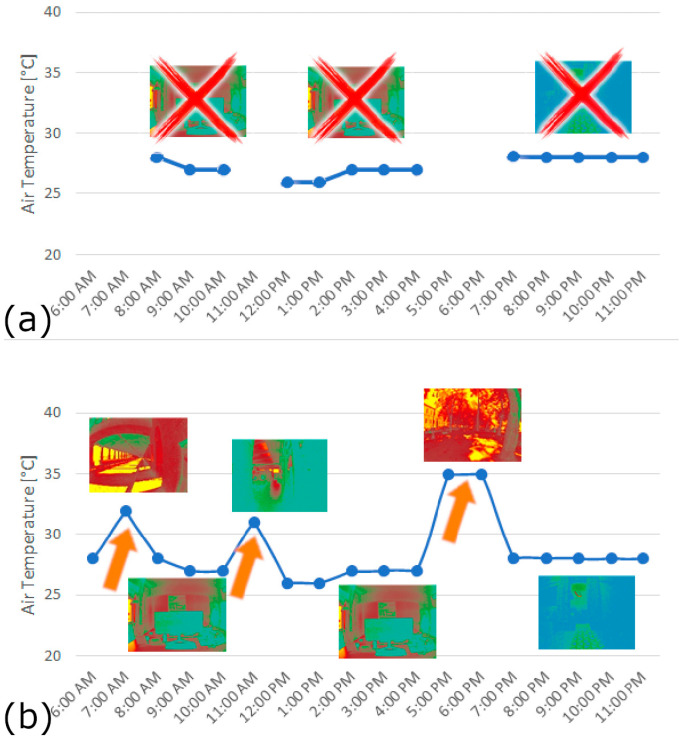
Example of air temperature data and luminance mapping: (**a**) with a nearable; (**b**) with a wearable.

**Table 1 sensors-24-06126-t001:** Users involved in the test: details about gender, test time, and use of the Empatica E4 wristband.

User	Gender	Day(s) of Test m = Morning, a = Afternoon	Empatica E4 Wristband Data
Anga Test 1	M	May 22 (m and a) May 23 (m and a) May 24 (m and a) May 29 (a)	No Yes Yes Yes
Anga Test 2	M	May 25 (m)	No
Anga Test 3	M	May 25 (a) May 26 (m)	No No
Anga Test 4	F	May 29 (m)	Yes
Anga Test 5	F	May 30 (m and a)	Yes

**Table 2 sensors-24-06126-t002:** Characteristics of sensors used for biometric data acquisition.

Sensor	Typical Range	Sampling Frequency
PPG sensor	-	64 Hz
EDA sensor	0.01 ÷ 100 µS	4 Hz
Skin Temperature sensor	−40 ÷ +85 °C	4 Hz
3-axes accelerometer	±2 g	32 Hz

**Table 3 sensors-24-06126-t003:** List of selected features (each selected feature is characterized by the symbol “•”).

#	Feature	List1 (267 Available Data for Each Feature)	List2 (403 Available Data for Each Feature)	List3 (403 Available Data for Each Feature)
1	In1_Out0	•	•	
2	CO2_ppm	•	•	•
3	PM1	•	•	
8	RH_8_%	•	•	
9	T_8_Â°C	•	•	
10	Va_m/s	•	•	
15	MRT [Â°C]	•	•	•
39	E_lx	•	•	
43	CCT	•	•	•
47	LAeq_R	•	•	•
59	HRV	•		
60	EDA	•		
61	BVP	•		
62	TEMP	•		
63	ACC_X	•		
64	ACC_Y	•		
65	ACC_Z	•		
66	ACC_Overall	•		
67	id_participant	•	•	
73	q_earphones	•	•	
74	q_visual_condition	•	•	
75	q_thermal_condition	•	•	•
76	q_air_quality_condition	•	•	
77	q_acoustic_condition	•	•	
78	q_general_comfort_condition	(target)	(target)	(target)

**Table 4 sensors-24-06126-t004:** Hyperparameters tuning range for different algorithms.

Algorithm	Hyperparameter	Range	Selected
RF	n_estimators	Range (1, 22, 2)	21
GBC	max_depth	Range (5, 16, 2)	7
	min_samples_split	Range (200, 1001, 200)	220
ETC	max_depth	Range (1, 50, 4)	29
	min_samples_leaf	[i/10.0 for i in range (1, 6)]	0.1
	max_features	[i/10.0 for i in range (1, 11)]	0.6
LSVC	penalty	[‘l1’, ‘l2’]	l2
	C	[100, 10, 1.0, 0.1, 0.01]	100
KNN	leaf_size	List (range (1, 50))	1
	n_neighbors	List (range (1, 30))	3
	p	[1, 2]	1
SVC	kernel	[‘poly’, ‘rbf’, ‘sigmoid’]	rbf
	C	[50, 10, 1.0, 0.1, 0.01]	50
	gamma	[‘auto’, ‘scale’, 1, 0.1, 0.01]	0.01

## Data Availability

The dataset generated and analysed during the current study is available from the corresponding author on reasonable request.

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
