# Peer review of "Integrated Approach for Human Wellbeing and Environmental Assessment Based on a Wearable IoT System: A Pilot Case Study in Singapore"

_sensors, 2024, doi:10.3390/s24186126_

Round 1

Reviewer 1 Report

Comments and Suggestions for Authors

The authors present a solution and a practical application developed for the evaluation of the occupants' perception of the environmental quality, in real conditions, by means of a multi-domain approach, considering thermal, visual, acoustic and air quality aspects, based on an IoT wearable system composed on a prototype of WEMoS-Wearable Environmental Monitoring System for environmental monitoring, a smart wristband for monitoring physiological data and a smart watch for obtaining subjective feedback from users. The integration and holistic interpretation of the multitude of combined data facilitate the prediction of user's overall comfort perception, considering the parameters of the environment in which the user is living/working, its physiological and subjective parameters.

The manuscript needs some improvements and aspects to be addressed before publication, as follows:

1.       Line 121: It must be “Section 4 the discussion and conclusions.”

2.       Line 140: For clarity, please specify if the study was done in an office/study building, not a residential one.

3.       Line 141: Please, also mention the approximate air volume of the indoor space.

4.       Line 150: In Table 1, for the clarity, please replace the column title “E4” with the title “Empatica E4 wristband data”.

5.       Line 181: For the sake of clarity, it will be important to show the images more clearly, to be readable, and/or the QR code larger in size.

6.       Line 201: In Table 2, please add a new column containing the accuracy of the specified sensors.

7.       Line 212: It must be “Thermohydrometers in the development of wearables. [24] have considered the distance to the”. Insert a dot before the square bracket [24]

8.       Lines 214-216: Please reformulate the text “A Senseair K30 carbon dioxide concentration sensor [32], an Adafruit PMSA003I particulate matter concentration sensor [33], considering different particle size: PM1, PM2.5 and PM10. A hot-wire anemometer from Modern device [34]”.

9.       Line 230: There is no mention about power supply solution. Please insert few words about it.

10.   Line 399: It must be “in real time”.

Author Response

Dear Reviewer, 

First of all, thank you for reviewing our manuscript and providing your valuable feedback.

Please consider the attached file, our responses are highlighted in yellow.

All changes in the revised manuscript are in track-changes mode.

Reviewer 2 Report

Comments and Suggestions for Authors

The paper introduces a novel methodology of using integrated wearable and environmental monitoring system as an alternative to the traditional method of measuring user satisfaction in a lab-controlled environemnt.

While this approach adds to the current field there are some points that need to be addressed. 

1-  The sample is 5 participants, would this affect the generalizability of the findings, would it be better to justify the sample size? 

2- The method of asking participants to reflect on their feedback since the last input might introduce recall bias (having participants recall what they have been feeling in the past). This is one of the challenges that micro EMA was designed to address. Could this potential bias affect the accuracy of the data collected and the subsequent analysis?

3-  The paper use multiple features for predictive analysis, but with a limited dataset, the complexity of the model might be disproportionate to the amount of data available. It would be helpful to discuss whether the sample size and the number of data points (403 non-zero values) are adequate to ensure model robustness and minimize the risk of overfitting. I understand that this is a pilot test, but addressing this would add clarity.

4- While the study introduces a comprehensive monitoring of environmental, physiological, and subjective comfort parameters, the discussion doesn’t include sufficient depth in evaluating how these diverse data sources interact as this is a major issue in the field.

5. It would be great to explore the broader implications of the findings, especially concerning the wearable system's application in real-world settings.

Other comments to improve the overall manuscript:

- Figure 1, please refer to the orignal figure where is is adapted from : https://www.sciencedirect.com/science/article/abs/pii/S0360132321009240
- Figure 5 text is too small
- Cozie App: Refer to the publication "Is your Clock face Cozie" instead of just the website.
- Mathematical symbol such as "T_1" would be better to be written in a mathematical formatting.

Author Response

(The authors gave the same response as above.)

Reviewer 3 Report

Comments and Suggestions for Authors

Dear Authors, I do not think there is too much to question about this paper. Clearly it is a pilot study, as clearly stated by the way. Thus, it should be considered as it is. Clearly the results are derived from "just" 5 participatants and that is a clear limitation. Another topic is that no comparison between the conventional models and this monitoring is provided. And another issue is referred to why that particular environment in Singapore?

I think that the addition of a "limitations of the study" paragraph will help to better define the research and its results, addressing what it is listed above

Author Response

Dear Reviewer, 

First of all, thank you for reviewing our manuscript and providing your valuable feedback.

In accordance with your suggestion, we have revised the article by dividing the Discussion and Conclusions into two separate paragraphs. In the new Discussion, we have added two subsections, 4.1 identifying potential limitations of our study and 4.2 describing broader implications related to the application of the wearable-based framework in real-world settings.

All changes in the revised manuscript are in track-changes mode.